# Research on the Energy Transfer Law of Polymer Gel Profile Control Flooding in Low-Permeability Oil Reservoirs

**DOI:** 10.3390/gels11070541

**Published:** 2025-07-11

**Authors:** Chen Wang, Yongquan Deng, Yunlong Liu, Gaocheng Li, Ping Yi, Bo Ma, Hui Gao

**Affiliations:** 1School of Petroleum Engineering, Xi’an Shiyou University, Xi’an 710065, China; cwangxsyu@163.com (C.W.); 15066439673@163.com (Y.D.); 15318794631@163.com (G.L.); 2Engineering Research Center of Development and Management for Low to Ultra-Low Permeability Oil & Gas Reservoirs in West China, Ministry of Education, Xi’an 710065, China; 3Key Laboratory of Exploration and Development of Complex and Difficult-to-Produce Oil & Gas Reservoirs, Ministry of Education, Xi’an 710065, China; 4Oil and Gas Technology Research Institute, PetroChina Changqing Oilfield Company, Xi’an 710018, China; 13236507857@163.com (Y.L.); 13776738856@163.com (P.Y.); 19992605070@163.com (B.M.); 5National Engineering Laboratory for Exploration and Development of Low Permeability Oil and Gas Fields, Xi’an 710018, China

**Keywords:** low-permeability oil reservoirs, polymer gel, profile control flooding, energy transfer law, injection methods, enhanced oil recovery

## Abstract

To investigate the energy conduction behavior of polymer gel profile control and flooding in low-permeability reservoirs, a parallel dual-tube displacement experiment was conducted to simulate reservoirs with different permeability ratios. Injection schemes included constant rates from 0.40 to 1.20 mL/min and dynamic injection from 1.20 to 0.40 mL/min. Pressure monitoring and shunt analysis were used to evaluate profile control and recovery performance. The results show that polymer gel preferentially enters high-permeability layers, transmitting pressure more rapidly than in low-permeability zones. At 1.20 mL/min, pressure onset at 90 cm in the high-permeability layer occurs earlier than in the low-permeability layer. Higher injection rates accelerate pressure buildup. At 0.80 mL/min, permeability contrast is minimized, achieving a 22.96% recovery rate in low-permeability layers. The combination effect of 1.2–0.4 mL/min is the best in dynamic injection, with the difference in shunt ratio of 9.6% and the recovery rate of low permeability layer increased to 31.23%. Polymer gel improves oil recovery by blocking high-permeability channels, expanding the swept volume, and utilizing viscoelastic properties.

## 1. Introduction

The proven reserves of low-permeability oil fields account for more than 60% of the domestic oil reserves, making them a crucial pillar of China’s petroleum industry [1,2,3]. However, low-permeability reservoirs face significant development challenges due to their complex physical properties and unique seepage characteristics. In recent years, the continuous development of these reservoirs has become an essential task. With the growing energy demand in China and the increasingly severe domestic energy security situation, the energy supply is facing enormous challenges. Meanwhile, the proportion of unconventional oil and gas production within the overall oil and gas output has been rising year by year. As an important unconventional oil and gas resource, low-permeability reservoirs cannot be overlooked [4,5,6,7,8].

Water injection, as an effective method for enhancing formation energy and crude oil production, is influenced by various factors such as water injection rates and injection pressure, which directly impact oil displacement efficiency [9,10]. To improve the development efficiency of low-permeability oil reservoirs, technologies like water injection and deep profile control flooding must be adopted. However, during the later stages of water injection development, particularly when the oil field enters a high water cut period, the effectiveness of water injection often falls short of expectations. For wells with a high water cut, deep profile modification technology is a commonly used solution. However, due to the unclear causal relationship between injection and production, the effectiveness of deep profile modification is often unsatisfactory. By optimizing and adjusting the displacement rate, it is hoped that the reservoir’s seepage structure can be improved, thereby enhancing the overall profile control flooding effect across the development interval or the entire reservoir.

Shen et al. developed a green and environmentally friendly bio-based profile control flooding agent, Bio Nano30, using non-covalent supramolecular interactions, specifically designed for ultra-low permeability oil reservoirs. Physical simulation experiments revealed the mechanism of profile control flooding with Bio Nano30. Both laboratory experiments and field applications demonstrated excellent results, including effective removal of wellbore blockages in oil wells, pressure reduction and enhanced injection in water wells, and successful profile modification and oil displacement in well groups [11]. Zhou et al. optimized the formulas of three deep profile control flooding gel systems for Block H of the Daqing Oilfield, a low-permeability reservoir, through orthogonal experiments. The profile control flooding effects of each system were further evaluated through core experiments [12]. Li et al. conducted compatibility experiments on profile control agents (PCAs) and dominant flow channels, which are key targets for deep profile modification agents in low-permeability reservoirs. They investigated three commonly used PCAs and three levels of dominant flow channels based on their permeability. Considering the permeability differences between layers, they proposed a systematic criterion for optimizing slug flow structures and engineering parameters through principal component analysis. This approach, based on the principles of “blocking, controlling, and displacing,” provides a valuable reference for PCA selection and field implementation parameters [13]. Zhao et al. employed polymer microspheres to enhance deep profile modification in low-permeability oil reservoirs. The plugging performance and deformability of the polymer microspheres improved with swelling time, making them effective pore channel plugging agents with strong deep profile modification capabilities in low-permeability cores [14]. Zou et al. introduced a new type of in situ polymer microsphere profile modification agent to resolve the issue of balancing injectability and profile modification effectiveness in low-permeability oil reservoirs. The in situ polymer microsphere (ISPM) emulsion is injected into the formation, where it undergoes polymerization to form microspheres for profile modification. Dual-layer heterogeneous core displacement experiments showed that ISPM can expand the swept volume of subsequent injected fluids, resulting in increased oil production and improved water control [15]. Abdullah et al. studied oil extraction in two-phase incompressible fluid in a two-dimensional rectangular porous uniform region filled with oil and without capillary pressure. By comparing three kinds of nanoparticles—SiO_2_, Al_2_O_3_, and CuO—the oil displacement process of nano-fluid and the influence of inlet temperature are simulated. The results show that adding nanoparticles into the base fluid can improve the oil recovery by more than 20% [16].

Li et al. selected three commonly used deep profile control agents (PCAs): nano-microsphere suspensions, polyethylene glycol single-phase gel particles (PEG), and cross-linked gel swelling particles (CBG-SP), and investigated the compatibility of these agents with fracture channels through a series of experiments. The experimental results demonstrated that nano-microspheres with particle sizes of 100 nm and 300 nm exhibit good injectability and deep migration ability, maintaining a relatively high core sealing rate [17]. Yin et al. combined the characteristics and evolution mechanisms of well pattern fractures, established a core model of outyield plate fractures considering fracture direction, and conducted core displacement experiments using the HPAM/Cr3+ gel deep conditioning system. They analyzed the mechanism of enhancing oil recovery by integrating deep profile control with circulating water injection. The results indicated that combining deep profile control with circulating water injection can be carried out simultaneously to further enhance recovery rates [18]. Xu et al. prepared preformed granular gel and optimized its synthesis conditions, investigating its temperature and salt resistance as well as its sealing performance. Field application tests were also conducted. The results revealed that PPGs possess good temperature and salt resistance. PPG with larger particle sizes exhibited greater sealing strength but a shorter effective period. Field tests showed that multiple rounds of profile control using PPGs of different particle sizes could achieve deep profile control [19]. By selecting two different types of reservoirs, Pryazhnikov et al. conducted oil displacement experiments with silica nanosol. The results show that the oil displacement efficiency increases with the increase in nanoparticle concentration and the decrease in nanoparticle size [20]. Yin et al. optimized the polymer/chromium ion deep profile control system, considering the characteristics of fractured and low-permeability reservoirs in the Chaoyanggou Oilfield, through viscosity evaluation, fluidity experiments, and oil displacement tests. The experimental results demonstrated that both high-molecular-weight main agent/low-concentration systems and low-molecular-weight main agent/high-concentration systems meet the required gel strength [21].

The Jurassic reservoirs of the Changqing Oilfield exhibit low permeability. After prolonged water flooding, dominant flow channels have become well-developed, causing injected water to rapidly advance along high-permeability layers. At the same time, the widespread development of edge and bottom water has led to a rapid rise in water cut in oil wells, resulting in significant oil production loss and a decline in the efficiency of conventional water injection development. Profile control flooding has emerged as one of the most promising enhanced oil recovery techniques, with successful industrial applications in oil fields such as Daqing and Shengli, achieving recovery rate increases of over 10% [22,23,24]. Previous research has highlighted that viscoelasticity is a critical property in profile control flooding. Polymer gel solutions improve the water-to-oil mobility ratio by enhancing the viscosity of the aqueous phase, thereby increasing the macroscopic swept volume. The elastic effects of these gels also improve microscopic oil displacement efficiency [25,26,27,28]. As a result, viscoelastic profile control flooding demonstrates superior oil displacement capabilities. Pilot tests in the Changqing Jurassic reservoirs have yielded positive results, stabilizing oil production and controlling water production. However, field observations have shown that, after polymer gel injection based on geological allocation, some wells exhibit both declining fluid levels and liquid production. This has created a pressing need to optimize the injection and production scheme for profile control flooding to boost formation energy around the production wells and achieve balanced liquid production. Due to the strong heterogeneity of low-permeability reservoirs, conventional injection methods and parameters for the main oil layers are no longer effective. Presently, there are few studies or reports addressing the adjustment of injection and production parameters during profile control flooding in such reservoirs.

This study is based on laboratory parallel double-tube displacement experiments. By utilizing an online real-time pressure monitoring system, the energy transfer dynamics during the profile control flooding process are quantitatively assessed. Additionally, the study investigates various combinations of injection rates to offer essential technical support for the efficient development of heterogeneous reservoirs.

## 2. Results and Discussion

### 2.1. Pressure Conduction Characteristics

After restoring the saturated crude oil in the sand-filled pipe model to its original oil–water distribution, water flooding was conducted at an injection rate of 0.50 mL/min. Figure 1 illustrates the pressure variation curves during the water flooding process for both the high-permeability and low-permeability models. Due to differences in permeability across the models and the injection method combining injection and separate extraction, the initial injection pressure of the low-permeability model is relatively high. The pressure in the high-permeability model rises sharply, eventually aligning with the inlet pressure of the low-permeability model. Following this, the pressure in both models decreases initially, then increases as the injection volume rises, eventually stabilizing [29,30,31]. When the injection volume reaches 1.40 PV, the pressure at each measurement point in the high-permeability sand-filled pipe reaches its lowest point and begins to stabilize. During the stage of oil pressure decline in the high-permeability sand-filled pipe, the pressure drop at each monitoring point remains relatively consistent. Once the injection volume hits 1.50 PV, the pressure at all measurement points in both the high-permeability and low-permeability sand-filled pipes stabilizes. The pressure in the low-permeability sand-filled pipe drops significantly at the 70 cm and 90 cm positions, closer to the outlet, during the early stage of water flooding. However, when the injection volume reaches 0.50 PV, the pressure at these points tends to stabilize. From the pressure curve, it can be observed that the pressure values remain generally low throughout the water flooding process. The maximum pressure in the high-permeability sand-filled pipe is 0.10 MPa, while the minimum pressure in the low-permeability sand-filled pipe is 0.10 MPa. The water flooding process ends when the cumulative moisture content at the outlet ends of both sand-filled pipes exceeds 90%. This phase simulates the situation on production sites where a large amount of water is produced at the extraction end after long-term water flooding.

After water flooding, a viscoelastic polymer gel solution is injected. The increased viscosity of the displacing medium leads to higher seepage resistance. Maintaining a constant injection rate inevitably causes the pressure to rise [32,33]. By monitoring pressure changes at different locations in high-permeability and low-permeability models, the specific flow direction of the gel at various injection volumes can be determined, and the pressure transfer characteristics during profile control flooding can be investigated.

During the profile control and flooding phase, injection rate significantly impacts pressure changes within the sand-filled pipe. As shown in the pressure change curve in Figure 2, within the injection rate range of 0.40 mL/min to 1.20 mL/min, polymer gel preferentially enters the high-permeability sand-filled pipe. The pressure transmission rate in the high-permeability pipe is significantly faster than in the low-permeability pipe, with all monitoring points showing pressure fluctuations, particularly near the inlet. At an injection rate of 0.40 mL/min (Figure 2a,b), the injection volume reaches 0.25 PV, and pressure in both sand-filled pipes begins to rise simultaneously at 10 cm. As injection progresses, the pressure at 90 cm in the high-permeability sand-filled pipe noticeably increases at 1.75 PV. However, the low-permeability sand-filled pipe does not experience pressure buildup until the injection volume reaches 3.10 PV at 90 cm. After injecting 2.70 PV and 3.10 PV, the pressures in both the high-permeability and low-permeability sand-filled pipes gradually stabilized. The maximum pressure at 10 cm in both pipes reached 6.50 MPa. When the injection rate is increased to 0.80 mL/min (Figure 2c,d), the pressure at the inlet end (10 cm) shows a noticeable increase after injecting 0.50 PV, indicating gel blockage at this point. However, the pressure in the low-permeability pipe does not begin to rise until the injection volume reaches 0.90 PV. At the outlet end (90 cm), the initial injection volumes for the high-permeability and low-permeability pipes are 1.50 PV and 2.80 PV, respectively. When the injection rate reaches 1.20 mL/min (Figure 2e,f), pressure in the high-permeability sand-filled pipe begins to rise at 10 cm with an injection volume of only 0.10 PV. In contrast, the low-permeability pipe only experiences pressure change when the injection volume reaches 0.20 PV, further confirming that the viscoelastic self-regulating agent preferentially enters the high-permeability channel. Compared to the initial pressure at 90 cm, the high-permeability sand-filled pipe exhibits an initial pressure of 1.49 PV, while the low-permeability sand-filled pipe shows 2.13 PV. This further demonstrates that pressure transmission is faster in the high-permeability pipe than in the low-permeability pipe.

Figure 3 shows the distributary ratio characteristics during the profile control and flooding phase at injection rates ranging from 0.40 to 1.20 mL/min. The distributary ratios in high- and low-permeability sand-filled pipes show both similarities and differences. At 0.40 mL/min (Figure 3a), the initial shunt rate of high-permeability pipes is 67.36%, roughly double that of low-permeability pipes (32.64%). As gel injection continues, high-permeability channels are blocked, reducing the diversion rate in high-permeability pipes and increasing it in low-permeability pipes, improving water injection balance [34,35,36]. When the injection volume reaches 2.40 PV, the diversion rates stabilize at 53.60% for high-permeability pipes and 46.40% for low-permeability pipes. At 0.80 mL/min (Figure 3b), the shunt rates change more rapidly. Initially, high-permeability pipes have a shunt rate of 88.09%, while low-permeability pipes have 11.90%. With continued injection, high-permeability channels are blocked, and the shunt ratio changes. By 2.50 PV, the diversion rates stabilize at 52.30% for high-permeability pipes and 47.70% for low-permeability pipes. At 1.20 mL/min (Figure 3c), the shunt rates initially differ significantly, with high-permeability pipes at 72.49% and low-permeability pipes at 27.51%. As gel injection progresses, high-permeability channels are blocked, and the shunt curves converge. However, the diversion rate curves reverse in the early mid-stage, suggesting that the gel’s blocking effect is limited. Ultimately, the diversion rates stabilize at 61.10% for high-permeability pipes and 38.90% for low-permeability pipes.

Due to the permeability grade differences, when the polymer gel solution is injected into a heterogeneous reservoir, it tends to flow more into the high-permeability regions, resulting in an increase in seepage resistance and a rapid rise in pressure. Consequently, the suction pressure difference in the low-permeability zones increases, causing the gel to start flowing towards these areas. As a result, the pressure rise in the low-permeability zones lags behind that in the high-permeability zones [37,38,39,40]. Additionally, the injection volume required for the outlet of the high-permeability region to begin pressurizing is less than that required for the low-permeability region, indicating that pressure conduction in the high-permeability model, dominated by the dominant channels, occurs more rapidly than in the low-permeability matrix.

During the experiment, pressure fluctuations were observed at each monitoring point, particularly near the injection end. This phenomenon can be attributed to the viscoelastic properties of the system. When the shear force is large, the physical associations are disrupted, leading to a decrease in viscosity and a drop in pressure. Conversely, when the shear force is small or removed, the association effect is restored, the structural viscosity is re-established, and the pressure increases.

During the constant rate profile control flooding process, differences in the split flow rates between the high-permeability and low-permeability models were observed at different stages. To reduce these differences and improve the recovery rate, the flow rate was adjusted during the displacement process, and a dynamic profile control flooding method was employed for the experiment.

Figure 4 illustrates the pressure change curves during the profile control flooding process with different multi-stage flow rates in the high-permeability and low-permeability models. As the injection volume of polymer gel increased, the pressure rose rapidly. The initial injection rate was 1.20 mL/min, which was reduced to 0.80 mL/min when the injection volume reached 2.10 PV. Figure 5 shows that at this point, the difference in split flow rates between the high-permeability and low-permeability models increased. To stabilize the split flow rate, the injection rate was adjusted to 0.80 mL/min, resulting in a significant drop in pressure at each monitoring point. As the injection volume continued to rise, the pressure gradually increased. When the injection volume reached 3.15 PV, the difference in split flow rates still increased, prompting a further reduction in flow rate to stabilize the split flow rate. From the split flow rate curve, it is evident that after the flow rate was reduced to 0.40 mL/min, the split flow rate gradually stabilized. When the injection volume reached 3.91 PV, the flow rate was further reduced to 0.10 mL/min in an attempt to reduce the split flow rate difference. However, the experimental data indicated that the difference could not be minimized. As a result, the displacement process was stopped when the injection volume reached 4.25 PV. Ultimately, the split flow rate in the high-permeability sand-packed tube was 70.40%, while the split flow rate in the low-permeability sand-packed tube was 32.60%.

As shown in Figure 6, the pressure increases gradually as the injection volume rises during profile control flooding. The initial injection rate was set at 0.80 mL/min, which was increased to 1.20 mL/min once the injection volume reached 2.10 PV. From Figure 7, it is evident that when the injection volume reached 2.10 PV, the split flow rate had already stabilized. As a result, the injection rate was adjusted to 1.20 mL/min to observe how the recovery rate changed with the variation in split flow rate. As the injection volume continued to increase, the pressure also rose steadily. By the time the injection volume reached 4.87 PV, the final split flow rate in the high-permeability model was 80.90%, while in the low-permeability model, it was 19.10%.

It can be observed from Figure 8 that as the injection volume of the polymer gel increases, the pressure rises rapidly. Initially, the injection rate was set at 1.20 mL/min. When the injection reached 1.50 PV, the injection was halted, and the flow rate was reduced to 0.40 mL/min. Figure 9 shows that when the injection volume reached 1.50 PV, the separation rate of the fracture hyperpermeability model approached 50% of that of the matrix hypopermeability model. Therefore, to maintain a better separation rate, the injection rate was reduced at this point. Afterward, the separation rate gradually diverged and eventually stabilized. In the end, the hyperpermeability separation rate was 54.80%, while the low permeability diversion rate was 45.20%.

### 2.2. Law of Energy Conduction

By comparing the pressure transfer characteristics under different conditions, the energy transfer law during viscoelastic profile control flooding in heterogeneous reservoirs was clarified. The stabilized pressure at each monitoring point was statistically analyzed, the pressure gradient between adjacent locations was calculated, and the pressure gradient distribution in different position segments was plotted.

As shown in Figure 10, the pressure gradient in the hypertonic model, with an injection rate of 0.80 mL/min, was highest in the 60–70 cm range, indicating the greatest accumulation of gel and the best sealing effect in this region. The pressure gradually decreased as it extended to both sides, forming a unimodal pattern. In the low-permeability model, the highest pressure gradient occurred in the 50–60 cm section, followed by the 10–20 cm section, with a 40 cm gap between them. These correspond to the main and secondary wave peaks, respectively, suggesting that the sealing effect follows a stepped pattern and is more effective.

The pressurization time at the production end under different conditions provides a relatively intuitive reflection of the pressure conduction rate in the sand-filling pipe. As shown in Figure 11, when the injection speed is increased under similar conditions in permeability ratio, the amount of polymer gel required for the end pressure of production in both high-permeability and low-permeability models decreases gradually, and the change in low permeability is more obvious. Compared with the injection speed of 0.40 mL/min, the amount of gel required for the end pressure of production decreases by 0.30 PV at 0.80 mL/min and by 0.97 PV at 1.20 mL/min, which indicates that the increase in injection speed makes the pressure low. Within a certain range, the energy conduction efficiency of the formation is positively correlated with the injection speed, meaning that a higher injection rate at the injection end leads to a faster pressure increase at the production end.

As shown in Figure 12, at an injection speed of 0.80 mL/min, the difference in the final diversion rate between the hypertonic and hypotonic models is the smallest, at only 4.60%. At this point, the instantaneous flow rates entering both models are nearly identical, indicating that the hypertonic channel is effectively blocked and the reservoir’s heterogeneity is significantly improved. When the injection rate is 1.20 mL/min, the high pressure gradient allows better utilization of the hypotonic layer in the early stage of injection. However, as the amount of retained gel increases, the suction pressure difference gradually grows. In the later stage, profile reversal begins, and the gel enters the hypertonic layer. The final diversion rate difference increases to 22.24%, and the effect deteriorates.

As shown in Figure 13, the difference in the final diversion rate between the hypertonic and hypotonic models at an injection rate of 1.20–0.40 mL/min is 9.60%. This relatively small difference indicates that the hypertonic channels are effectively blocked, significantly improving the heterogeneity of the reservoir. When the injection speeds were 1.20–0.80–0.40–0.10 mL/min, the difference in the final diversion rate between the hypertonic and hypotonic models increased to 37.80%. At an injection speed of 0.80–1.20 mL/min, the difference in the diversion rate rose to 61.72%. If the difference in the diversion rate is too large, most of the fluid will flow through the hypertonic crack model, resulting in poor overall effectiveness.

### 2.3. Recovery Rate Characteristics

Before conducting the static profile control flooding experiments, water flooding experiments were initially performed. During the water flooding process, a combined injection and separate production method was employed. The water flooding was halted once the cumulative water cut at the outlet of both the high-permeability model and the matrix model exceeded 90%. Following this, profile control flooding experiments were conducted with different constant injection rates. The results demonstrated that the recovery rate improved to varying extents after the profile control flooding. The experimental data is presented in Table 1.

The injection rate for water flooding in all three experimental groups was set at 0.5 mL/min. As shown in Figure 14, the recovery rate during the water flooding stage was approximately 76% for the high-permeability model and around 68% for the low-permeability model, demonstrating good repeatability of the experiments. When comparing the changes in recovery rate after profile control flooding, the increase in recovery rate for both the high-permeability and low-permeability models was smallest at an injection rate of 0.40 mL/min. In contrast, at an injection rate of 1.20 mL/min, the recovery rate for the low-permeability model increased by 2.65%, reaching 17.25%. The most significant increase in recovery rate for the low-permeability model occurred at an injection rate of 0.8 mL/min, where it rose by 8.36% compared to 0.4 mL/min, reaching 22.96%. This improvement is largely attributed to the optimal split flow rate achieved at this injection rate. At 0.80 mL/min, the difference in the final split flow rate between the high-permeability and low-permeability models was smallest, at only 4.57%, with nearly identical instantaneous flow rates entering both models. This indicates that the high-permeability channels were effectively blocked, leading to a significant improvement in the reservoir’s heterogeneity. At an injection rate of 1.20 mL/min, the higher pressure gradient allowed better utilization of the low-permeability layer during the early stage of injection. However, as the amount of retained gel increased, the suction pressure difference gradually escalated, resulting in profile reversal in the later stages, with the gel entering the high-permeability layer. This led to a final difference in split flow rate of 22.24%, resulting in a less effective outcome.

Through dynamic profile control flooding experiments, the changes in recovery rate under multi-stage flow rates were explored, and the optimal injection parameter experimental data are shown in Table 2.

The injection rate for water flooding in all three experimental groups was set at 0.50 mL/min. As shown in Figure 15, the recovery rate during the water flooding stage was approximately 65.50% for the high-permeability model and around 64.50% for the low-permeability model, indicating a good repeatability of the experiments. When comparing the changes in recovery rate after profile control flooding, the smallest increase in recovery rate for both the high-permeability and low-permeability models occurred at injection rates of 1.20, 0.80, 0.40, and 0.10 mL/min. In contrast, at an injection rate of 0.80–1.20 mL/min, the recovery rate for the low-permeability model increased by 3.75%, reaching 17.81%. The most significant improvement in recovery rate for the low-permeability model was observed at an injection rate of 1.20–0.40 mL/min, where it increased by 17.17% compared to the 1.20–0.80–0.40–0.10 mL/min injection rate, reaching 31.23%. This change is closely related to the optimal split flow rate achieved at this injection rate. Specifically, at an injection rate of 1.20–0.40 mL/min, the difference in the final split flow rate between the high-permeability and low-permeability models was the smallest at only 9.64%, with nearly identical instantaneous flow rates entering both models. This indicates that the high-permeability channels were effectively blocked, leading to a significant improvement in the reservoir’s heterogeneity.

### 2.4. Microscopic Mechanism of Profile Control Flooding

Profile control flooding technology aims to enhance reservoir heterogeneity and improve crude oil recovery rates, with its microscopic mechanisms being particularly critical [41,42]. In heterogeneous reservoirs, profile control agents play a vital role by preferentially entering larger pores with higher permeability. These agents form high-strength plugging barriers through processes such as adsorption, bridging, and cross-linking, forcing the injected fluid to alter its flow direction. As a result, the fluid gradually moves into low-permeability areas, expanding the swept volume and effectively mobilizing oil from low-permeability reservoirs [43,44,45]. The ratio of G’/G’’ of polymer gel is 5.81, and the elastic modulus is greater than the viscous modulus, so the gel can better resist shear deformation. Moreover, due to its viscoelasticity, the residual oil flows at the pore throat by relying on the extrusion swelling effect, which significantly improves the oil recovery (Figure 16 and Figure 17) [46,47,48,49].

## 3. Conclusions

Based on sand-packed tube physical model experiments, a study on polymer gel profile control flooding was conducted to address the challenges of combined injection and separate production in high-permeability and low-permeability channels. The research explored the laws of formation energy transfer and the microscopic mechanisms during the water injection stage of polymer gel profile control flooding under different injection and production conditions. The following conclusions were drawn:(1)After polymer gel is injected into the heterogeneous reservoir, it preferentially enters the high-permeability layer, causing an increase in seepage resistance and a rapid rise in pressure. As the suction pressure differential increases, the gel gradually shifts towards the low-permeability layer, resulting in a delayed pressure response in the low-permeability layer. The pressure transfer speed in high-permeability channels is significantly faster than in low-permeability channels.(2)The formation energy transfer efficiency increases with the injection rate. The higher the injection rate, the faster the pressure increases at the production end, leading to a more significant reduction in the amount of polymer gel required to buildup pressure in the low-permeability reservoir layer.(3)Initially using a high flow rate to promote rapid pressure transfer to the low-permeability area, followed by a reduction in flow rate to maintain a stable split flow rate, can effectively block the high-permeability channels and significantly improve the recovery rate.(4)When the difference in split flow rate between the high-permeability and low-permeability reservoir layers is minimized, the heterogeneity of the reservoir is most effectively improved, resulting in the largest increase in recovery rate.(5)By blocking the high-permeability channels, polymer gel causes the fluid to gradually shift towards the low-permeability reservoir, thereby expanding the swept volume. Moreover, due to its viscoelasticity, the polymer gel utilizes the extrusion swelling effect to overcome capillary forces, causing the residual oil in the pore throats to flow and thus increasing the crude oil recovery rate.

## 4. Materials and Methods

### 4.1. Materials and Instruments

The quartz sand ground from the Yan 9 outcrop of Hujianshan Oilfield has a particle size of 40–60 mesh (0.3–0.5 mm). The degassed crude oil of Hujianshan Oilfield has a viscosity of 1.50 mPa·s at 50 °C. In terms of water types, the simulated formation water is a CaCl_2_ water type with a mineralization degree of 52,197 mg/L; the simulated injected water is a Na_2_SO_4_ water type with a mineralization degree of 1470 mg/L. Polymer gel comprised 1000 mg/L of the mixture (this was produced by Xi’an Changqing Chemical Industry Group Co., Ltd. Xi’an, China. It was analytically pure, with the elastic modulus of the system greater than the viscosity modulus, and the viscosity remaining at low shear stress and decreasing at high shear, showing good viscoelastic characteristics. It has good thermal stability, and thermogravimetric analysis shows that the fastest range of weight loss rate is 320–400 °C. Its salt resistance is outstanding; the viscosity is 6.1 mPa·s in saline water with a salinity of 50,000 mg/L, and the viscosity retention rate was about 75%).

Figure 18 shows a schematic diagram of the indoor parallel double-tube displacement experimental device, which consists of four key components: (1) Heterogeneous reservoir simulation system: Two parallel sand-filling pipes are used to simulate the reservoir matrix and high-permeability bands, respectively, with combined injection and separate production (sand-filling pipe, length × diameter = 1000 mm × 50 mm, 9 pressure monitoring points, set at equal intervals of 100 mm). (2) Fluid injection system: 1 L intermediate container, 2PB-104IV type constant flow pump; (3) Online real-time pressure monitoring system: A total of 18 pressure monitoring points, which can record the pressure changes at different parts of the sand-filling pipe from the injection end to the production end in real time. (4) Oil–water metering system: Liquid collection bottle.

### 4.2. Experimental Principle

Convert the injection speed based on the on-site injection linear speed [50], and use Equation (1) to approximate the injection speed for the large-scale object model replacement experiment. This will allow for the setting of the experimental parameters, as outlined in Table 3 and Table 4.(1)$$Q=\phi AV_{a}=\frac{\pi \phi D^{2}V_{a}\times 100}{4\times 24\times 60}$$

Of these, Q is the injection speed of the experiment, ml/min; ϕ is porosity, %; A is the cross-sectional area of the core, cm^2^; $V_{a}$ is the linear velocity at the time of injection in the mine, m/d; and D represents the diameter of the core, in cm.

The diversion rate is defined as the ratio of the liquid production from high-permeability (or low-permeability) cores to the total liquid production. During the displacement process, the liquid output from both high and low permeability models is measured in real-time, and their respective diversion rates are calculated using Equations (2) and (3), as shown in Sections [51,52,53,54].(2)$$D_{h}=\frac{V_{h}}{V_{h}+V_{l}}\times 100\%$$(3)$$D_{l}=\frac{V_{l}}{V_{h}+V_{l}}\times 100\%$$

Of these, *D_h_* represents the diversion rate of the hypertonic model, %; *D_l_* represents the diversion rate of the hypotonic model, in %; *V_h_* represents the liquid production volume of the hypertonic model, in ml; *V_l_* represents the liquid production volume of the hypotonic model, in mL.

### 4.3. Experimental Procedures

Based on parallel double-tube displacement experiments in the laboratory, combined with an online real-time pressure monitoring system, the energy transfer law during the profile control flooding process was quantitatively evaluated. The specific steps of the physical flow simulation experiments for profile control flooding in the sand-packed tube are as follows:(1)Considering the characteristics of the reservoir, such as strong heterogeneity and significant oil–water separation in the Yan’an Formation, a mixture of quartz sand and clay was prepared for sand packing in tubes with different permeabilities, in accordance with the experimental requirements.(2)The sand-packed tubes were filled with the prepared sand mixture, and the weight of each tube was recorded.(3)The intelligent pressure monitoring system was installed, and the airtightness of the device connections was thoroughly checked.(4)The two sand-packed tubes were saturated with formation water at a flow rate of 0.5 mL/min. The permeability was measured using the liquid, and after complete saturation, the sand-packed tubes were weighed to calculate their pore volume.(5)The two sand-packed tubes were then saturated with crude oil at a flow rate of 0.5 mL/min. Saturation was halted when the liquid at the outlet reached 100% crude oil.(6)The sand-packed tubes were connected in parallel, and water flooding experiments were conducted at a flow rate of 0.5 mL/min. The displacement was stopped when the cumulative water cut of the two sand-packed tubes reached 90%. During the water flooding process, pressure changes at different positions of the sand-packed tubes were monitored in real time. Additionally, the water cut and oil displacement efficiency of the two sand-packed tubes were recorded every half hour.(7)After completing the water flooding experiment, the viscoelastic self-regulating agent displacement experiment was carried out at the set experimental flow rate. Throughout the experiment, pressure changes at various positions of the sand-packed tubes were monitored and recorded in real time, with the water cut and oil displacement efficiency of the tubes also recorded every half hour.(8)The displacement was stopped when the injected volume reached five times the total pore volume of the two sand-packed tubes, or when the water cut at the outlet of the sand-packed tubes remained unchanged.(9)After completing the displacement experiment, the packing material in the sand-packed tubes was removed, the tube walls were cleaned, and after complete drying, the permeability contrast of the sand-packed tubes was reassessed. Alternatively, adjustments were made to the injection rate of the profile control flooding agent, or the timing of rate changes during dynamic profile control flooding. Steps 2 to 8 were then repeated as necessary.

## Figures and Tables

**Figure 1 gels-11-00541-f001:**
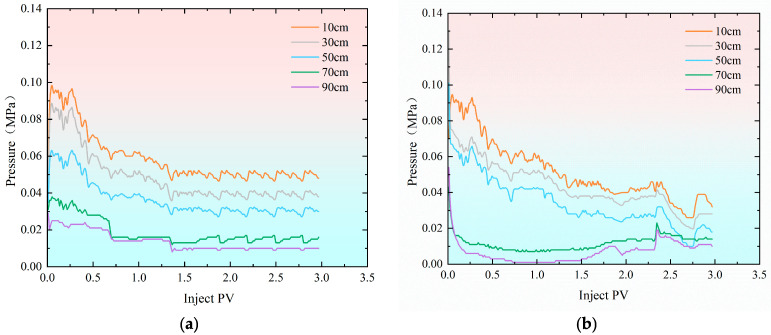
Pressure variation curve during the water flooding stage. (**a**) Hypertonic model; (**b**) hypotonic model.

**Figure 2 gels-11-00541-f002:**
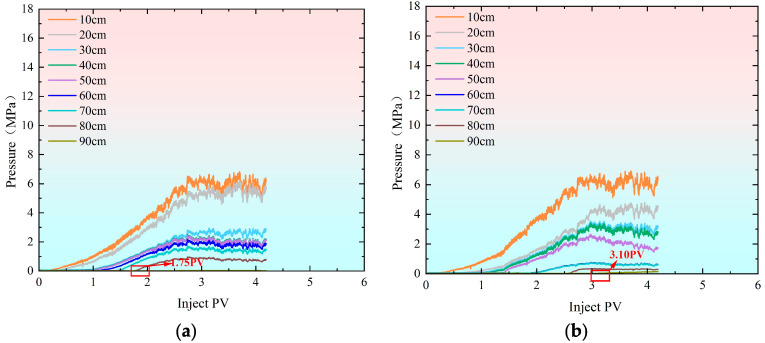
Pressure change curve at injection rate of 0.40 mL/min–1.20 mL during profile control and flooding. (**a**) Hypertonic model (injection rate: 0.40 mL/min); (**b**) hypotonic model (injection rate: 0.40 mL/min). (**c**) Hypertonic model (injection rate: 0.80 mL/min); (**d**) hypotonic model (injection rate: 0.80 mL/min). (**e**) Hypertonic model (injection rate: 1.20 mL/min); (**f**) hypotonic model (injection rate: 1.20 mL/min).

**Figure 3 gels-11-00541-f003:**
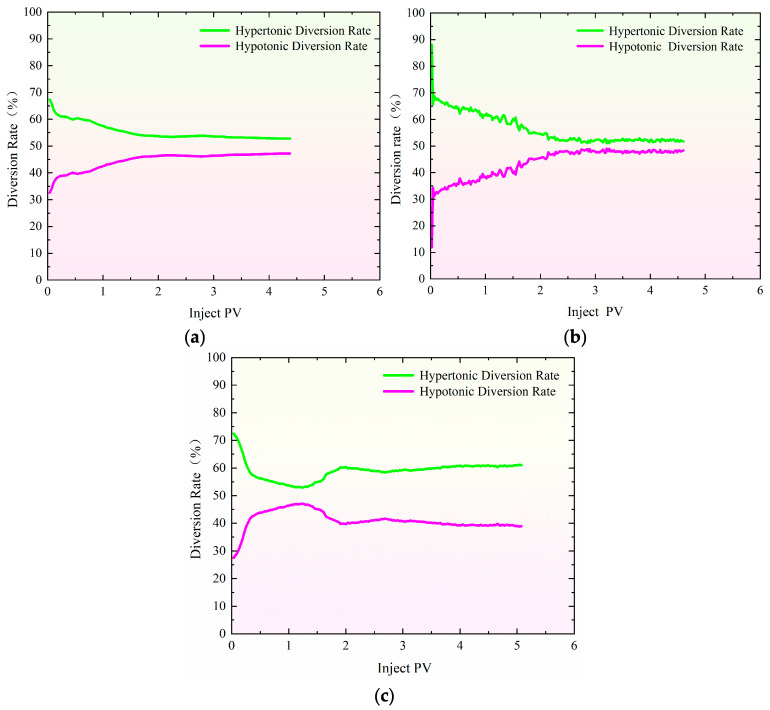
Special evidence of profile control and flooding shunt rate curve with injection rate of 0.40 mL/min–1.20 mL/min. (**a**) Injection rate: 0.40 mL/min; (**b**) injection rate: 0.80 mL/min; (**c**) injection rate: 1.20 mL/min.

**Figure 4 gels-11-00541-f004:**
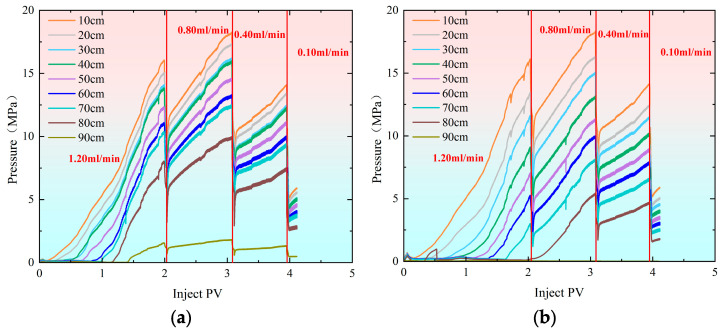
Pressure change curves during profile control flooding at injection rates of 1.20–0.80–0.40–0.10 mL/min. (**a**) Hypertonic model (injection rate: 1.20–0.80–0.40–0.10 mL/min); (**b**) hypotonic model (injection rate: 1.20–0.80–0.40–0.10 mL/min).

**Figure 5 gels-11-00541-f005:**
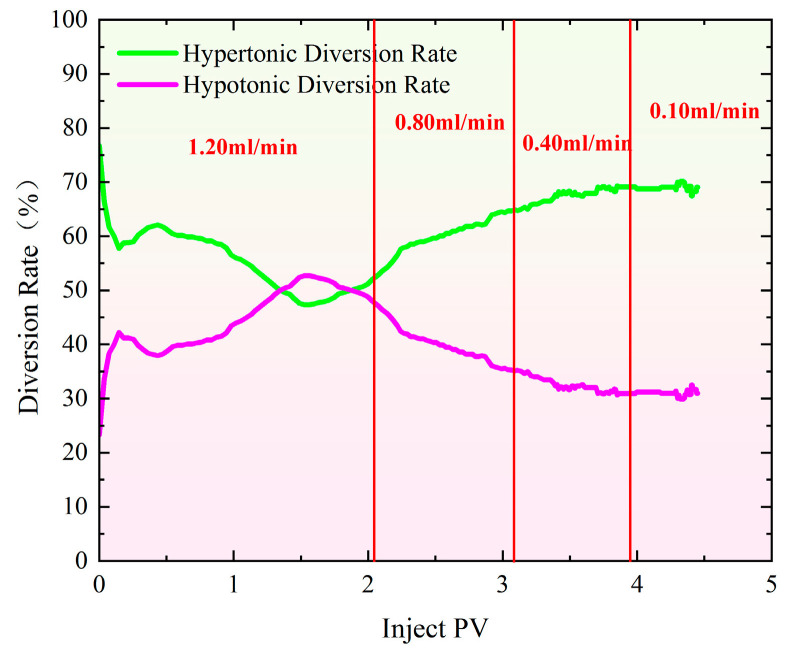
Characteristics of split flow rate curves during profile control flooding at injection rates of 1.20–0.80–0.40–0.10 mL/min.

**Figure 6 gels-11-00541-f006:**
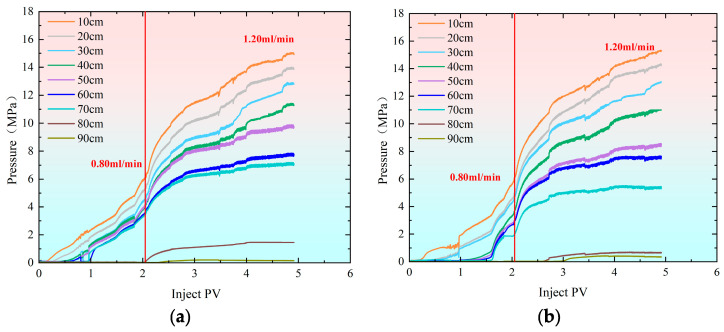
Pressure change curves during profile control flooding at injection rates of 0.80–1.20 mL/min. (**a**) Hypertonic model (injection rate: 0.80–1.20 mL/min); (**b**) hypotonic model (injection rate: 0.80–1.20 mL/min).

**Figure 7 gels-11-00541-f007:**
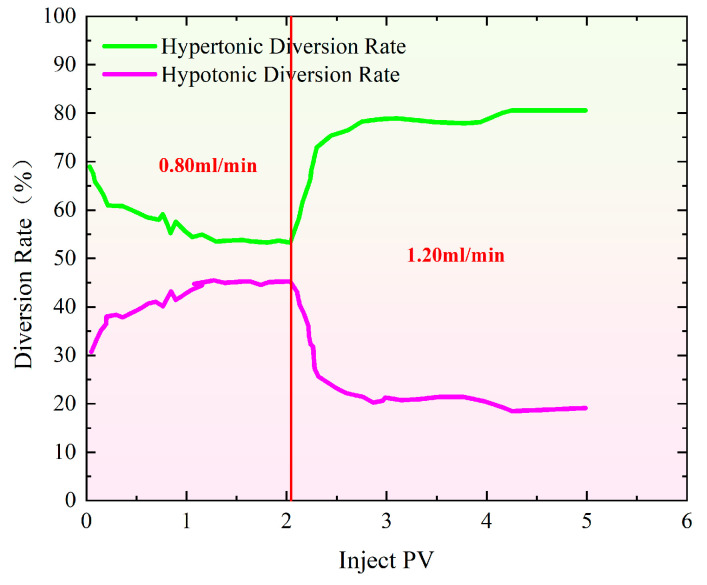
Characteristics of split flow rate curves during profile control flooding at injection rates of 0.80–1.20 mL/min.

**Figure 8 gels-11-00541-f008:**
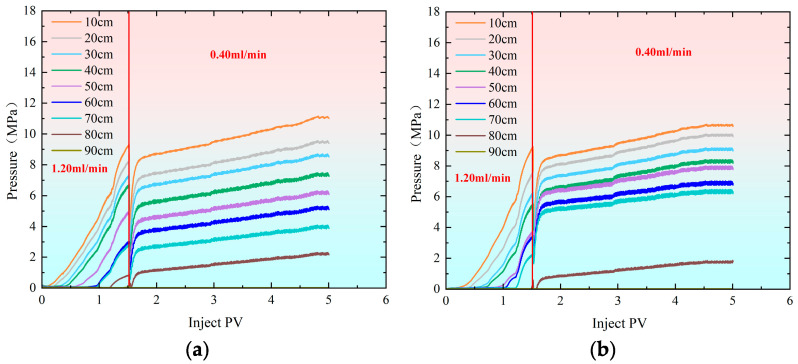
Pressure change curves during profile control flooding at injection rates of 1.20–0.40 mL/min. (**a**) Hypertonic model (injection rate: 1.20–0.40 mL/min); (**b**) hypotonic model (injection rate: 1.20 mL/min).

**Figure 9 gels-11-00541-f009:**
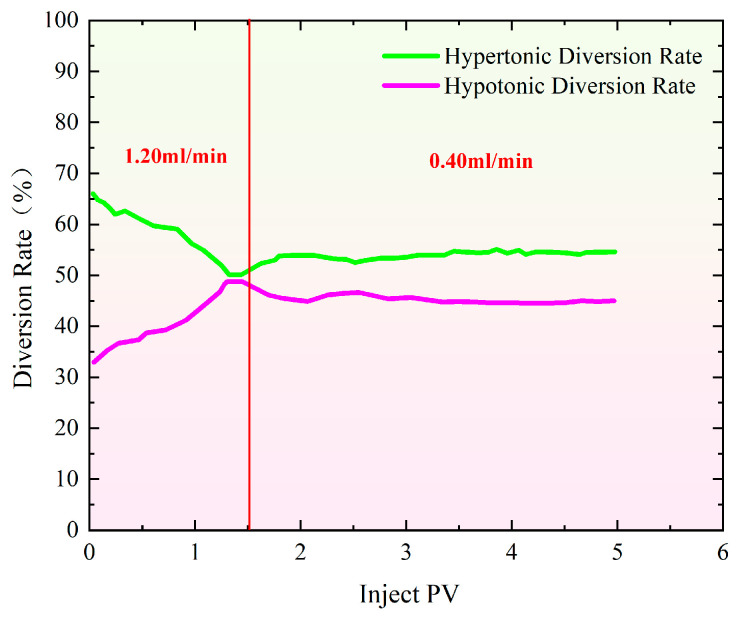
Characteristics of split flow rate curves during profile control flooding at injection rates of 1.20–0.40 mL/min.

**Figure 10 gels-11-00541-f010:**
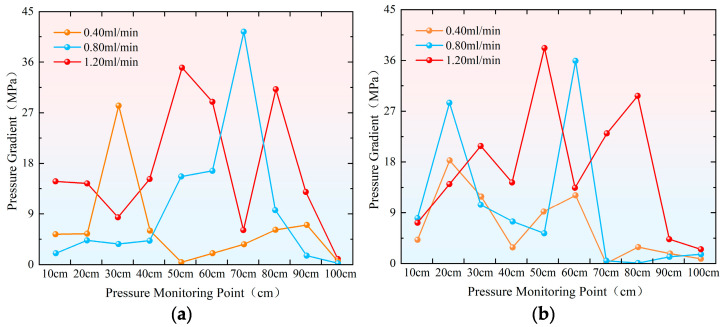
Shows the line graph of the average pressure gradient distribution at each position of the sand-filling pipe under different injection speeds. (**a**) Hypertonic model; (**b**) hypotonic model.

**Figure 11 gels-11-00541-f011:**
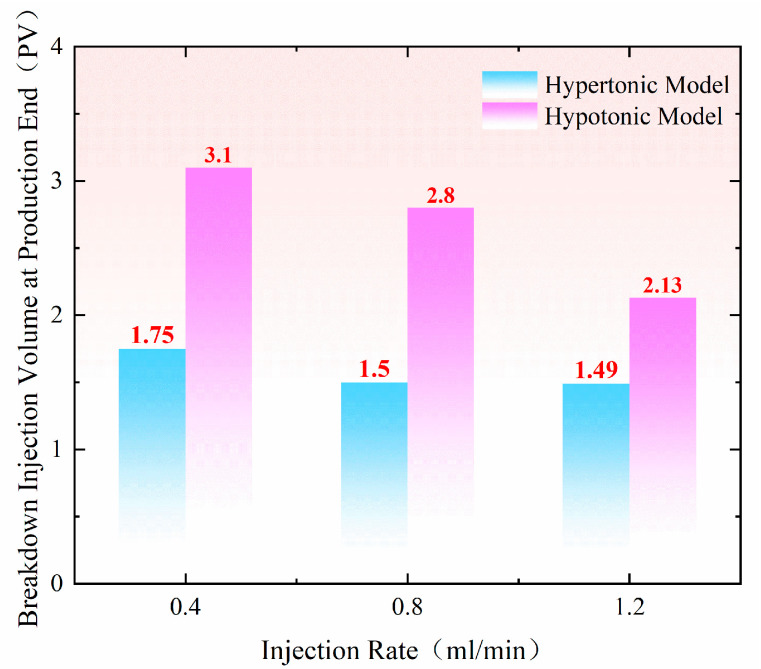
Shows the distribution of pressure buildup time at the production end under different constant injection speeds.

**Figure 12 gels-11-00541-f012:**
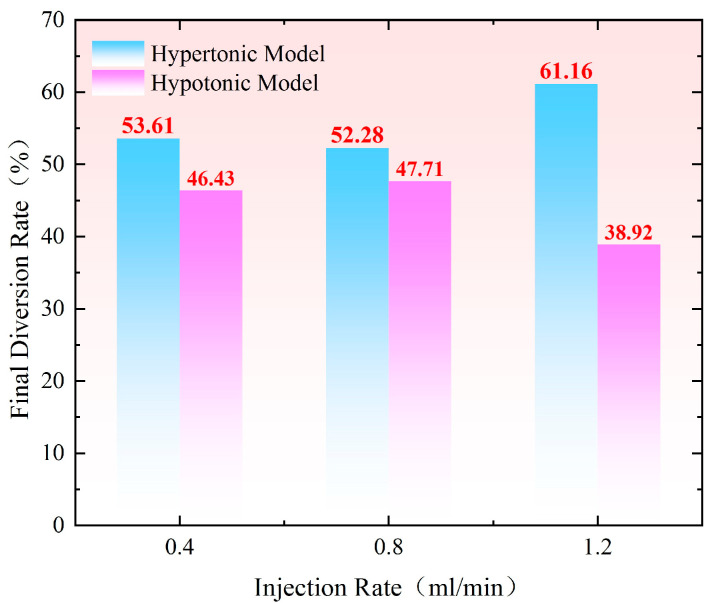
Comparison of the final diversion rate under different constant injection speeds.

**Figure 13 gels-11-00541-f013:**
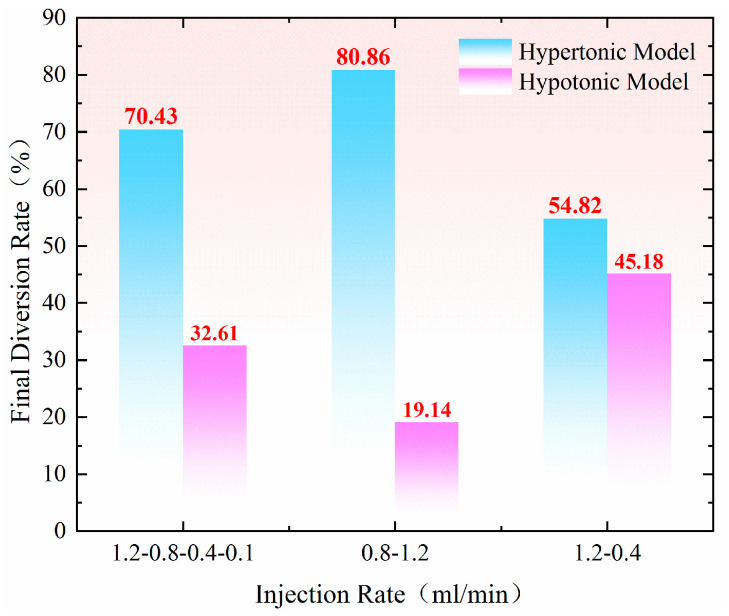
Comparison of final diversion rates under multi–pole flow rates.

**Figure 14 gels-11-00541-f014:**
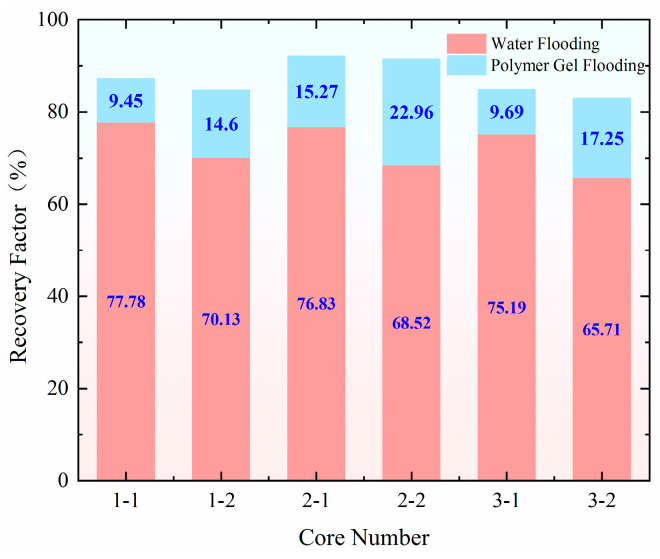
Comparison of the extraction degrees at each stage of the static displacement experiment.

**Figure 15 gels-11-00541-f015:**
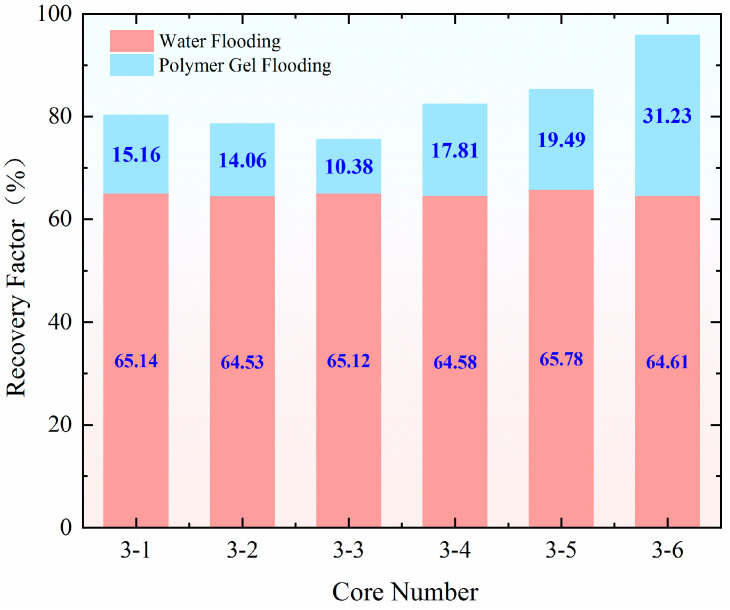
Comparison of the extraction degrees at each stage of the dynamic displacement experiment.

**Figure 16 gels-11-00541-f016:**
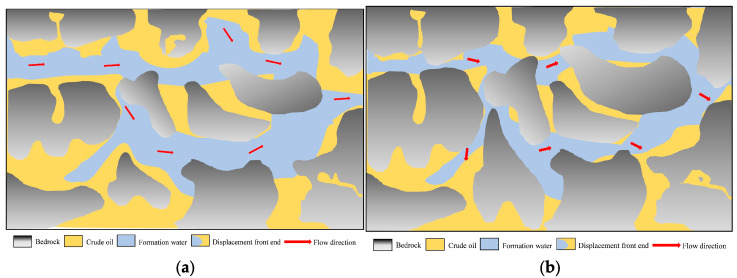
Microscopic mechanism of water flooding. (**a**) Water flooding state of the hypertonic model; (**b**) water flooding state of the low-permeability model.

**Figure 17 gels-11-00541-f017:**
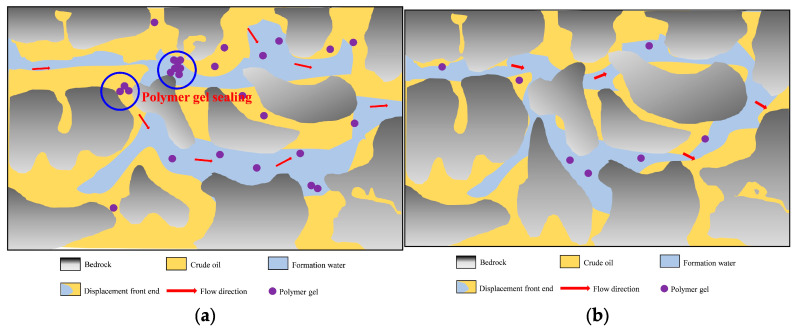
Microscopic mechanism of polymer gel profile control flooding. (**a**) Profile control flooding state of high-permeability model; (**b**) profile control flooding state of low-permeability model.

**Figure 18 gels-11-00541-f018:**
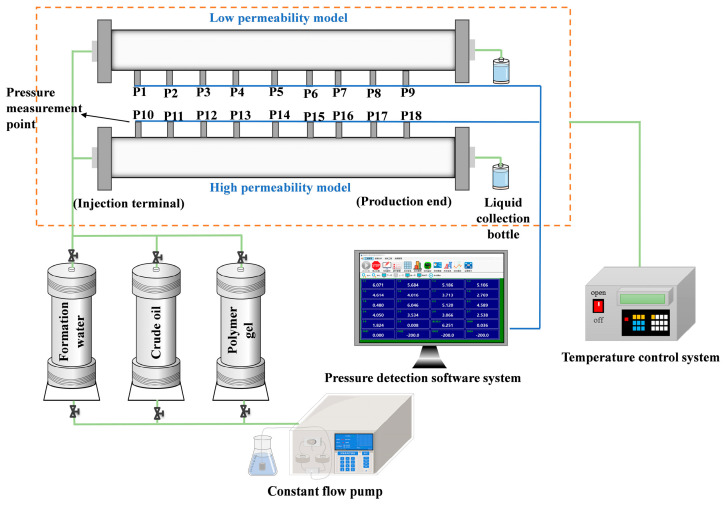
Schematic diagram of the indoor parallel double-tube displacement experimental device.

**Table 1 gels-11-00541-t001:** Recovery rate data for water flooding and profile control flooding.

Core Number	Water Flooding Flow Velocity (mL/min)	Profile Control Flooding Rate (mL/min)	Permeability (mD)	Water Flood Recovery Rate (%)	Profile Control Flooding Recovery Rate (%)	Total Recovery Rate (%)
1-1	0.50	0.40	107.30	77.78	9.45	87.23
1-2			54.20	70.13	14.60	84.73
2-1	0.50	0.80	101.00	76.83	15.27	92.10
2-2			47.90	68.52	22.96	91.48
3-1	0.50	1.20	100.60	75.19	9.69	84.88
3-2			47.05	65.71	17.25	82.96

**Table 2 gels-11-00541-t002:** Comparison of recovery rates at multiple flow rates.

Core Number	Water Flooding Flow Velocity (mL/min)	Profile Control Flooding Rate (mL/min)	Permeability (mD)	Water Flooding Recovery Rate (%)	Profile Control Flooding Recovery Rate (%)	Total Recovery Rate (%)
3-1	0.50	1.20–0.80–0.04–0.10	101.00	65.14	15.16	80.30
3-2			48.00	64.53	14.06	78.59
3-3	0.50	0.80–1.20	106.90	65.12	10.38	75.5
3-4			49.50	64.58	17.81	82.39
3-5	1.20–0.40	1.20–0.40	103.80	65.78	19.49	85.27
3-6			49.40	64.61	31.23	95.84

**Table 3 gels-11-00541-t003:** Comparison between on-site injection linear velocity and experimental injection velocity.

Method	On-Site Injection Linear Velocity (m/d)	Experimental Injection Rate (ml/min)
Water flooding	0.013	0.50
Profile control and flooding	0.01	0.40
	0.02	0.80
	0.03	1.20

**Table 4 gels-11-00541-t004:** Design of displacement experiment parameters.

Experiment Number	Experimental Content	Core Number	Penetration Rate (mD)	The Penetration Rate Is Extremely Poor	Porosity (%)	Polymer Gel Concentration (mg/L)	Injection Speed (mL/min)	
							Water Flooding	Profile Control Flooding
1	Static Displacement	1-1	107.30	2.00	37.20	1000	0.50	0.40
		1-2	54.20		23.90			
2		2-1	101.00	2.10	34.50			0.80
		2-2	47.90		26.50			
3		3-1	100.60	2.10	33.40			1.20
		3-2	47.10		29.20			
4	Dynamic Displacement	4-1	101.00	2.10	29.20			1.20–0.80–0.40–0.10
		4-2	48.00		26.50			
5		5-1	106.90	2.20	29.30			0.80–1.20
		5-2	49.50		26.60			
6		6-1	103.80	2.10	29.50			1.20–0.40
		6-2	49.40		26.40			

## Data Availability

All data are included in the manuscript.

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
