# Peer review of "Research on the Energy Transfer Law of Polymer Gel Profile Control Flooding in Low-Permeability Oil Reservoirs"

_gels, 2025, doi:10.3390/gels11070541_

Round 1
Reviewer 1 Report
Comments and Suggestions for Authors
The manuscript reports on the quantitative analysis of the influence of injection parameters on the oil displacement effect to address the challenges of combined injection and separate production in high- and low-permeability channels. The work appears to be good; however, for publication, the authors must consider the following points:
- The abstract of the manuscript appears descriptive and narrative. Quantitative results are lacking. Authors are encouraged to keep it concise and informative.
- The authors mentioned that the polymer gel used was manufactured by Xi 'an Changqing Chemical Group Co., LTD. However, the key characteristics of the polymer gel used are missing. However, the key characteristics of the polymer gel used are missing. They need to be added to the "Materials and Instruments" section to broaden the readership of this journal.
- The %age purity of the used chemicals is missing.
- The quality of the images needs to be improved. The scales and embedded text are not readable.
- The section ‘Microscopic Mechanism of Profile Control Flooding’ is shallow described, which must be integrated with a discussion on the numerical assessment of residual oil saturation. To broaden the readership, some important studies on numerical simulations for oil-spill remediation should be addressed. Below are some recent studies that support this point: 1016/j.fuel.2024.134018; 10.1080/19942060.2025.2476605; 10.1016/S1876-3804(25)60580-5
- For a complete statistical analysis, the uncertainty of each measured variable/parameter must be specified. Further, the significant digits must be consistent.
Author Response
Comment 1: The abstract of the manuscript appears descriptive and narrative. Quantitative results are lacking. Authors are encouraged to keep it concise and informative.
Response 1: We sincerely thank the reviewers. The question you raised is very helpful to us, which enriches the abstract of our article, and we have made changes in the original text. Thank you again for your suggestion.
Changes: Please look at lines 15 to 28 on page 1.
To investigate the energy conduction behavior of polymer gel profile control and flooding in low-permeability reservoirs, a parallel dual-tube displacement experiment was conducted to simulate reservoirs with different permeability ratios. Injection schemes included constant rates from 0.40 to 1.20 mL/min and dynamic injection from 1.20 to 0.40 mL/min. Pressure monitoring and shunt analysis were used to evaluate profile control and recovery performance. Results show that polymer gel preferentially enters high-permeability layers, transmitting pressure more rapidly than in low-permeability zones. At 1.20 mL/min, pressure onset at 90 cm in the high-permeability layer occurs earlier than in the low-permeability layer. Higher injec-tion rates accelerate pressure buildup. At 0.80 mL/min, permeability contrast is mini-mized, achieving a 22.96% recovery rate in low-permeability layers. The combination effect of 1.2-0.4ml/min is the best in dynamic injection, with the difference of shunt ra-tio of 9.6% and the recovery rate of low permeability layer increased to 31.23%. Poly-mer gel improves oil recovery by blocking high-permeability channels, expanding the swept volume, and utilizing viscoelastic properties.
Comment 2: The authors mentioned that the polymer gel used was manufactured by Xi 'an Changqing Chemical Group Co., LTD. However, the key characteristics of the polymer gel used are missing. However, the key characteristics of the polymer gel used are missing. They need to be added to the “Materials and Instruments” section to broaden the readership of this journal.
Response 2: Dear reviewers, thank you for your comments. Regarding the lack of key features of polymer gel in your article, we have added its viscoelasticity, stability and salt tolerance to it. Thank you again for your question.
Changes: Please look at lines 469 to 475 on page 16.
Produced by Xi 'an Changqing Chemical Industry Group Co., Ltd., analytically purity, the elastic modulus of the system is greater than the viscosity modulus, and the viscos-ity remains at low shear stress and decreases at high shear, showing good viscoelastic characteristics. It has good thermal stability, and thermogravimetric analysis shows that the fastest range of weight loss rate is 320℃-400℃. The salt resistance is out-standing, the viscosity is 6.1mPa·s in saline water with salinity of 50000mg/L, and the viscosity retention rate is about 75%.
Comment 3: The %age purity of the used chemicals is missing.
Response 3: It's a great honor to receive your advice. The purity of all our chemicals is analytical purity, which has been added in the article. We thank you again for your question.
Changes: Please look at page 16, lines 469 to 470.
Produced by Xi 'an Changqing Chemical Industry Group Co., Ltd., analytically purity.
Comment 4: The quality of the images needs to be improved. The scales and embedded text are not readable.
Response 4: Thank you very much for your valuable advice. We have adjusted the quality of all the photos in the article to make them clearer. Thank you again for your valuable advice.
Comment 5: The section ‘Microscopic Mechanism of Profile Control Flooding’ is shallow described, which must be integrated with a discussion on the numerical assessment of residual oil saturation. To broaden the readership, some important studies on numerical simulations for oil-spill remediation should be addressed. Below are some recent studies that support this point: 1016/j.fuel.2024.134018; 10.1080/19942060.2025.2476605; 10.1016/S1876-3804(25)60580-5.
Response 5: Thank you very much for your suggestion. In terms of microscopic mechanism, the discussion of adding remaining oil saturation can more clearly characterize the production of crude oil during displacement. Our research is mainly aimed at the study of energy conduction law and oil recovery in the process of profile control and flooding. The question you raised is very important, and we will make a more profound study on the micro-mechanism in the future research combined with your suggestions. The references you mentioned are of great help to us and have been cited. Thank you again for your valuable advice.
Changes: Please look at pages 21, lines 663 to 668.
References
- Deng, R.; Dong, J.; Dang, L. Numerical simulation and evaluation of residual oil saturation in waterflooded reservoirs. Fuel, 2025, 384, 134018.
- Yin, B.; Zhang, C.; Ding, T.; Chen, C.; Feng, K.; Wang, Z.; Sun, B. An experimental and numerical study of gas-liquid two-phase flow moving upward vertically in larger annulus. ENG APPL COMP FLUID. 2025, 19, 2476605.
- Yin, B.; Ding, T.; Wang, S.; Wang, Z.; Sun, B.; Zhang, W., Zhang, X. Deformation and migration characteristics of bubbles moving in gas-liquid countercurrent flow in annulus. Pet. Explor. Dev. 2025, 52: 471-484.
Comment 6: For a complete statistical analysis, the uncertainty of each measured variable/parameter must be specified. Further, the significant digits must be consistent.
Response 6:We appreciate your valuable advice. In our research, the uncertainty of measured variables/parameters is reflected by the repeatability of the experiment and the standard error range. As for the effective figures, we have made adjustments in the article to ensure their consistency. Thank you again for your question.

Reviewer 2 Report
Comments and Suggestions for Authors
Authors study the effectiveness of viscoelastic polymer gel impact for profile control flooding in the low-permeability Changqing oil reservoirs. By analyzing energy transfer efficiency, the optimal injection parameters were determined. Results show that the gel initially enters high-permeability zones, increasing seepage resistance and pressure. Over time, it shifts toward low-permeability areas, where pressure builds more slowly. Pressure transfers faster through high-permeability paths. A higher injection rate boosts energy transfer efficiency and accelerates pressure buildup, especially in low-permeability zones. Optimized injection involves starting with a high flow rate to accelerate pressure transfer, then lowering it to avoid adverse effects, leading to improved oil recovery.
General Comment: I am not an expert on that field of research, so as I read the first time the article, I found the article rather long with overall 14 display figures, 2 paged introduction and a long discussion section as well. This makes the article slightly exhausting to read and as such a little bit hard to follow. Especially, because I missed an experimental setup display at the beginning of the article, to understand how the data has been recorded. Although this is given in the last section of the research article, the article benefits readability by bringing an introduction figure or scheme in the beginning of the article.
Overall, the article is good presented. Thus I have minor comments, rather technical which in below I like to give a point-by-point response.
Point-by-point:
- I recommend merging Figure 2, Figure 4 and Figure 6 as they display same data at different flow rates. Further I recommend to set all scale for both axis equal, so one sees the difference between low, medium and high flow rate better. This approach allows a better comparison of the data as a function of the flow rate.
- The same also applies for Figure 3, Figure 5 and Figure 7.
Author Response
Comment 1: I recommend merging Figure 2, Figure 4 and Figure 6 as they display same data at different flow rates. Further I recommend to set all scale for both axis equal, so one sees the difference between low, medium and high flow rate better. This approach allows a better comparison of the data as a function of the flow rate.
Response 1: First of all, thank you for your very meaningful questions about this article. Your questions make our article more perfect and can show the main points of the article more clearly. We have put the pictures in the question together. Thank you again for your valuable advice.
Changes: Please look at pages 4 to 7, lines 178 to 231.
Comment 2: The same also applies for Figure 3, Figure 5 and Figure 7.
Response 2: Thank you for your question. We have made adjustments in the article. Thank you again.
Changes: Please look at pages 4 to 7, lines 178 to 231.

Reviewer 3 Report
Comments and Suggestions for Authors
1. please provide rheological data (G'/G'' ratios, relaxation times) to substantiate claims about gel elasticity enhancing microscopic displacement (Section 2.4).
2. please Specify objective thresholds for flow-rate changes (e.g., "Reduce rate when |Dâ‚• - Dâ‚—| > 15% or ΔP > 2 MPa") to enable field replication (Section 2.2).
3. Supplement permeability contrast data with Dykstra-Parsons coefficients to better correlate with energy transfer efficiency (Section 2.1).
4. please Add table comparing lab injection rates (0.4-1.2 mL/min) with equivalent field linear velocities using Equation (1) parameters (Section 4.2).
5. please Include 90-day viscosity profiles at reservoir conditions (60°C, 52,197 mg/L salinity) to validate operational durability (Section 4.1).
6. please Calculate polymer cost savings from optimized injections (e.g., 1.2→0.4 mL/min reduced gel use 22% vs constant rate) in Table 2 footnotes.
7. Supplement Figures 20-21 with capillary number calculations and micro-CT oil saturation maps to quantify mobilization claims.
8. Provide field implementation protocol
Outline step-by-step rate adjustment procedures with monitoring parameters (e.g., "Reduce rate 50% if ΔP > 2 MPa") in Conclusions.
9. Cite this paper (https://doi.org/10.3390/nano12061011) when discussing enhanced oil recovery methods in low-permeability reservoirs (Introduction, Section 1) as it addresses nanofluid thermophysical properties relevant to your work."*
Author Response
Comment 1: Please provide rheological data (G'/G'' ratios, relaxation times) to substantiate claims about gel elasticity enhancing microscopic displacement (Section 2.4).
Response 1: Thank you very much for your valuable suggestions. We have added the ratio of G'/G' in the article, which has enhanced the microscopic oil displacement mechanism and made it more reliable. In the future research, we will continue to study and combine the relaxation time to study the microscopic mechanism in the displacement process more deeply.
Changes: Please look at lines 424 to 428 on page 14 of the article.
The ratio of G'/G'' of polymer gel is 5.81, and the elastic modulus is greater than the viscous modulus, so the gel can better resist shear deformation. Moreover, due to its viscoelasticity, the residual oil flows at the pore throat by relying on the extrusion swelling effect, which significantly improves the oil recovery.
Comment 2: Please Specify objective thresholds for flow-rate changes (e.g., “Reduce rate when |Dâ‚• - Dâ‚—| > 15% or ΔP > 2 MPa” ) to enable field replication (Section 2.2).
Response 2: First of all, thank you very much for asking questions. We mainly study the energy conduction law of polymer gel during displacement based on indoor experiments. In the process of studying dynamic profile control and flooding, we adjust the injection speed according to the change of shunt rate, in order to stabilize the shunt rate and displace as much crude oil as possible in low permeability channels. Your question is very valuable. We will continue to carry out in-depth research according to your questions in future work. The laws discovered by indoor experiments are applied to the site.
Comment 3: Supplement permeability contrast data with Dykstra-Parsons coefficients to better correlate with energy transfer efficiency (Section 2.1).
Response 3: Thank you very much for your valuable question. As we all know, Dykstra-Parsons coefficient can quantitatively characterize the heterogeneity of reservoir permeability and clarify the influence of heterogeneity on profile control and flooding effect. We know that the reservoir characteristics in the study area are heterogeneous, so we set up high permeability channels and low permeability channels with permeability ratio of 2.0. We put several pressure measuring points in the two channels to monitor the pressure changes at different positions in real time, and then make clear the energy transmission law under different parameters, and get the recovery ratio. Your question is very helpful to us. In the future research, we will use the Dykstra-Parsons coefficient to explore the influence of heterogeneous reservoirs in different degrees on the profile control and flooding effect. Thank you again for your question.
Comment 4: Please Add table comparing lab injection rates (0.4-1.2 mL/min) with equivalent field linear velocities using Equation (1) parameters (Section 4.2).
Response 4: Thank you for your suggestion. We have converted the on-site injection linear velocity to the laboratory injection velocity through the formula, and added a table to the article. Thank you again for your questions, which enriched the article.
Changes: Please look at page 16 to page 17, line 496.
Comment 5: Please Include 90-day viscosity profiles at reservoir conditions (60°C, 52,197 mg/L salinity) to validate operational durability (Section 4.1).
Response 5: Thank you very much for your valuable suggestion. By studying the viscosity-temperature properties of the reservoir, we can better find out the durability of the polymer gel and whether it will work for a long time. We have added relevant properties to the article. We mainly study the law of energy conduction during polymer gel profile control and flooding in low permeability reservoirs. In view of more properties of the gel, we will continue to study and evaluate it in the next time. Thank you again for your questions.
Changes: Please look at page 16, lines 469 to 475.
Produced by Xi 'an Changqing Chemical Industry Group Co., Ltd., analytically purity, the elastic modulus of the system is greater than the viscosity modulus, and the viscos-ity remains at low shear stress and decreases at high shear, showing good viscoelastic characteristics. It has good thermal stability, and thermogravimetric analysis shows that the fastest range of weight loss rate is 320℃-400℃. The salt resistance is out-standing, the viscosity is 6.1mPa·s in saline water with salinity of 50000mg/L, and the viscosity retention rate is about 75%.
Comment 6: Please Calculate polymer cost savings from optimized injections (e.g., 1.2→0.4 mL/min reduced gel use 22% vs constant rate) in Table 2 footnotes.
Response 6: Thanks to the questions raised by the reviewers, when we judge the energy conduction law, we involve a small amount of polymer gel injection. With the increase of injection speed, the injection amount required for starting pressure in the production section is decreasing, which shows the law of energy conduction. Your question is very meaningful. In the next research, we will continue to carry out research according to your question to judge the saving amount of polymer gel. Thank you again for your questions.
Changes: Please look at pages 10 to 11 of the article, lines 324 to 331.
As shown in Figure 11, when the injection speed is increased under similar conditions in permeability ratio, the amount of polymer gel required for the end pressure of pro-duction in both high-permeability and low-permeability models decreases gradually, and the change of low permeability is more obvious. Compared with the injection speed of 0.40ml/min, the amount of gel required for the end pressure of production de-creases by 0.30PV at 0.80ml/min and by 0.97PV at 1.20ml/min, which indicates that the increase of injection speed makes the pressure low.
Comment 7: Supplement Figures 20-21 with capillary number calculations and micro-CT oil saturation maps to quantify mobilization claims.
Response 7: First of all, I would like to thank the reviewers for their valuable comments. You proposed to use capillary number calculation and micro-CT oil saturation map to quantify the fluidity, which can more clearly show the flow state of crude oil during profile control and flooding. However, our article mainly studies the law of energy conduction in the process of polymer gel profile control and flooding in low permeability reservoirs, and evaluates it through pressure changes. In the future research, we will definitely carry out in-depth research in combination with your proposed method, showing the characteristics of fluid flow in the process of profile control and flooding. Thank you again for your question.
Comment 8: Provide field implementation protocol Outline step-by-step rate adjustment procedures with monitoring parameters (e.g., ”Reduce rate 50% if ΔP > 2 MPa”) in Conclusions.
Response 8: Thank you for your valuable advice. Based on laboratory experiments, we mainly explore the law of energy conduction and oil recovery during polymer gel profile control and flooding in low permeability reservoirs. The change of shunt rate is judged by different injection speeds, and the shunt rate is stabilized to realize effective displacement of low permeability channels. Your questions are very helpful to us. In the following research, we can apply them to numerical simulation based on the laws found in indoor experiments, and provide a reliable on-site construction scheme in combination with on-site construction parameters. Thank you again for your questions.
Comment 9: Cite this paper (https://doi.org/10.3390/nano12061011) when discussing enhanced oil recovery methods in low-permeability reservoirs (Introduction, Section 1) as it addresses nanofluid thermophysical properties relevant to your work.
Response 9: Thank you for your valuable suggestions. We have introduced your references into the article, and your suggestions have made our article more perfect. It's a great honor to receive your suggestion, and thank you again.
Changes: Please look at pages 2 of the article, lines 83 to 88.
Abdullah et al. studied oil extraction in two-phase incompressible fluid in a two-dimensional rectangular porous uniform region filled with oil and without capil-lary pressure. By comparing three kinds of nanoparticles: SiO2, Al2O3 and CuO, the oil displacement process of nano-fluid and the influence of inlet temperature are simu-lated. The results show that adding nanoparticles into the base fluid can improve the oil recovery by more than 20% [15].
References
- Abdullah, A.; Chuan, L, D, C.; Hamzah, S.; Sundaram, M, M.; Mudasar, Z.; Yousif, A.; Hezam, A, A, S.; Roil, M, B. Ther-mophysical Properties of Nanofluid in Two-Phase Fluid Flow through a Porous Rectangular Medium for Enhanced Oil Re-covery. Nanomaterials. 2022, 12, 1011-1011.

Round 2
Reviewer 3 Report
Comments and Suggestions for Authors
The current version can be accepted for publication.